# The controlled disassembly of mesostructured perovskites as an avenue to fabricating high performance nanohybrid catalysts

Yuan Wang[1], Hamidreza Arandiyan[1], Hassan A. Tahini[2], Jason Scott[1], Xin Tan[2], Hongxing Dai[3], Julian D. Gale[4], Andrew L. Rohl[4], Sean C. Smith[2] & Rose Amal[1]

Versatile superstructures composed of nanoparticles have recently been prepared using various disassembly methods. However, little information is known on how the structural disassembly influences the catalytic performance of the materials. Here we show how the disassembly of an ordered porous $La_{0.6}Sr_{0.4}MnO_3$ perovskite array, to give hexapod mesostructured nanoparticles, exposes a new crystal facet which is more active for catalytic methane combustion. On fragmenting three-dimensionally ordered macroporous (3DOM) structures in a controlled manner, via a process that has been likened to retrosynthesis, hexapod-shaped building blocks can be harvested which possess a mesostructured architecture. The hexapod-shaped perovskite catalyst exhibits excellent low temperature methane oxidation activity ($T_{90\%} = 438\,^\circ C$; reaction rate $= 4.84 \times 10^{-7}\,mol\,m^{-2}\,s^{-1}$). First principle calculations suggest the fractures, which occur at weak joints within the 3DOM architecture, afford a large area of (001) surface that displays a reduced energy barrier for hydrogen abstraction, thereby facilitating methane oxidation.

[1] Particles and Catalysis Research Group, School of Chemical Engineering, The University of New South Wales, Sydney, New South Wales 2052, Australia. [2] Integrated Materials Design Centre (IMDC), School of Chemical Engineering, The University of New South Wales, Sydney, New South Wales 2052, Australia. [3] College of Environmental and Energy Engineering, Beijing Key Laboratory for Green Catalysis and Separation, and Laboratory of Catalysis Chemistry and Nanoscience, Beijing University of Technology, Beijing 100124, China. [4] Department of Chemistry, Curtin Institute for Computation, Curtin University, PO Box U1987, Perth, Western Australia 6845, Australia. Correspondence and requests for materials should be addressed to H.A. (email: h.arandiyan@unsw.edu.au) or to S.C.S. (email: sean.smith@unsw.edu.au) or to R.A. (email: r.amal@unsw.edu.au).

It is of great interest to use methane as an alternative fuel to coal and oil due to its high ratio of hydrogen to carbon which leads to comparatively lower greenhouse gas emissions. Catalytic methane oxidation is often employed to stabilize 'lean' flames (those with a low fuel to air ratio) at relatively low temperatures as compared to non-catalytic combustion, thus preventing the formation of noxious nitrogen oxides[1]. Precious metals (for example, Pd and Pt) supported on $Al_2O_3$ are well studied and used as commercial catalysts for complete methane oxidation at low temperatures. However, the associated high cost and poor thermal stability (caused by agglomeration of the metal deposits) of the catalytic elements persist as major challenges. Mixed oxides comprising non-noble metal components seem a more promising alternative, with perovskites having gained particular attention[2,3]. Using a perovskite as a catalyst for methane combustion can improve stability although this is offset by a partial sacrifice in activity. Efforts to minimize the associated activity loss have focused on using perovskite make-up and preparation techniques to influence factors such as element composition, morphology, surface area and structure control. It is commonly accepted that scaling down particle dimensions to the nanometre regime can generate unique properties within the material[4]. For example, nanosized particles can significantly increase the surface area and thus increase the contact between reactants and surface active sites[5]. In prior work[6], three-dimensionally ordered macroporous $La_{0.6}Sr_{0.4}MnO_3$ (3DOM LSMO) was synthesized with a high surface area and a rich presence of surface oxygen species whereby it exhibited a much better catalytic performance than its one-dimensional (1D) counterpart. Recently, new nanoparticle shaping strategies have been established involving the disassembly of preformed macroscopic architectures by cutting, pasting or merging assembled structures. For example, Fukino et al.[7] investigated hollow nanotube arrays which were assembled from ferrocene-based tetratopic pyridyl ligands mixed with $AgBF_4$. Upon oxidizing the ferrocene groups, the tubes could be cut into a large, stable rings and then reversibly reassembled into nanotubes by reduction of the ferrocene groups. The findings reported by Fukino et al.[7] opened the door to many attractive new possibilities with the approach also suggesting a new protocol towards mixed crystal nanoparticle synthesis.

Herein, we implement a rational fragmentation strategy involving a 3DOM architecture as the precursor material, which is disassembled into well-defined structural building units to produce a mesostructured LSMO perovskite catalyst. The disassembly approach is promising for acquiring nanostructures for a wide range of catalytic applications which are otherwise difficult to obtain by other templating methods. We demonstrate the synthesis of 3D-hm LSMO, where h refers to hexapod and m refers to mesostructured. The hexapod building blocks are derived from a highly crystalline mesoporous 3DOM framework, which is an inverse replica of a well-ordered hard template polymethyl methacrylate (PMMA) microsphere array. The surface chemistry and structure of the hexapod perovskite particles were examined by X-ray photoelectron spectroscopy (XPS) and density functional theory (DFT) computational studies so as to understand the origin of the enhanced methane oxidation activity exhibited by the 3D-hm LSMO.

## Results

**Synthesis and structural characterizations**. A 3D hexapod mesoporous LSMO perovskite structure was synthesized using a sequential assembly/disassembly strategy. An interconnected solid skeleton with a 3DOM structure was initially fabricated by infiltrating the interstitial spacing between a well-ordered polymer microsphere array with metal precursors (nitrates of lanthanum, strontium and manganite) after which the polymer microsphere template was thermally removed (Supplementary Methods). The 3DOM LSMO comprises face-centered cubic arrays of micrometre-scale cavities and 2D hexagonal close-packed arrays of nanometre channels. The 3DOM LSMO structure was then fragmented using ultrasound so as to break the weak connection points and generate the close-packed mesostructured 3D-hm LSMO. The resulting 3D-hm LSMO consists of hexapod building blocks with sizes between $5-40$ nm as depicted by the 3D model in Fig. 1. For comparison, a 1D nonporous (1DDN) LSMO sample was also prepared using a citrate complexing method[8]. The synthesis parameters for the LSMO samples are summarized in Supplementary Table 2. The as-obtained LSMO samples were assessed for methane oxidation. Before activity testing the samples were oxidized in pure $O_2$ (30 ml per min) at 300 °C for 2 h. Fragmenting the 3DOM framework in a controlled manner was found to be important with the structures comprising a low degree of connectivity points between structural building blocks that were prone to cleavage under specific conditions (Supplementary Figs 3 and 4).

An assessment of the structural evolution of PMMA, the 3DOM network, and the 3D-hm LSMO was conducted using field-emission high-resolution scanning electron microscopic (FE-HRSEM) and 3D-environmental atomic force microscopic (3D-eAFM) techniques (Fig. 2). Figure 2a,b provide images of the ordered PMMA microspheres (average microsphere diameter $\sim 210$ nm), showing an interconnected sphere porosity exists between the hexagonal microsphere arrangement. Greater than 98% of the particles identified in the FE-HRSEM images display a high-quality 3DOM structure with a high degree of alignment perpendicular to the first layer (Fig. 2c,d). The 3DOM LSMO comprises face-centered cubic arrays of micrometer-scale cavities and 2D hexagonal close-packed arrays of nanometre channels. Shrinkage of the close-packed periodic voids during the 3DOM LSMO calcination step results in the spherical void size ($\sim 140$ nm) being $25-35\%$ smaller than the initial PMMA microsphere diameter (Supplementary Fig. 6). The low degree of connectivity within the 3DOM LSMO mesostructure enables its disassembly into unique building blocks upon sonication treatment. The ultrasound treatment sees the formerly curved pore interfaces flatten out and form hexapod-like building blocks (Fig. 2e,f) with a body diameter of $\sim 80$ nm and leg lengths of $\sim 15$ nm. The hexapod building blocks appear to be randomly packed to form reticular mesopores with channel diameters between $5-40$ nm, as shown in Fig. 2f. The mesopore channels have an open framework with accessible pores and no obvious aggregation of the building blocks is observed. The elemental distribution and composition of the as-synthesized 3DOM LSMO were auxiliary-mapped using silicon drift detector energy dispersive spectrometry (EDS) by evaluating the integrated intensity of the La, Sr, Mn and O signals as a function of the beam position when operating the FE-HRSEM/EDS in scanning mode (Supplementary Fig. 7). During the synthesis of 3D-hm LSMO materials, the critical process is the fracture of the continuously interconnected skeleton. Breakage at the weak connection points can be realized by internal or external forces (such as dissolution or sonication, respectively). Compared to the dissolution strategy, sonication generates better mesostructures as the dissolution process leads to particle aggregation, which may diminish the benefits of the nanoscopic sizes as well as block the mesopores.

The field-emission high-resolution transmission electron microscopy (FE-HRTEM) technique can provide more complete and compelling detail on the structure and particle morphology. The spherical elements of self-assembled monodisperse PMMA

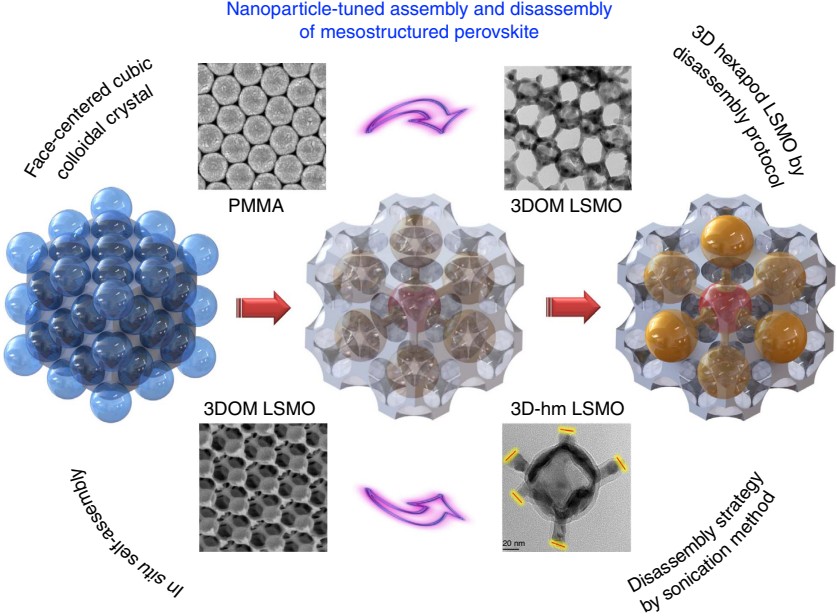

**Figure 1 | Synthesis steps of 3D-hm LSMO.** Schematic illustration depicting synthesis of the 3D-hm LSMO catalysts.

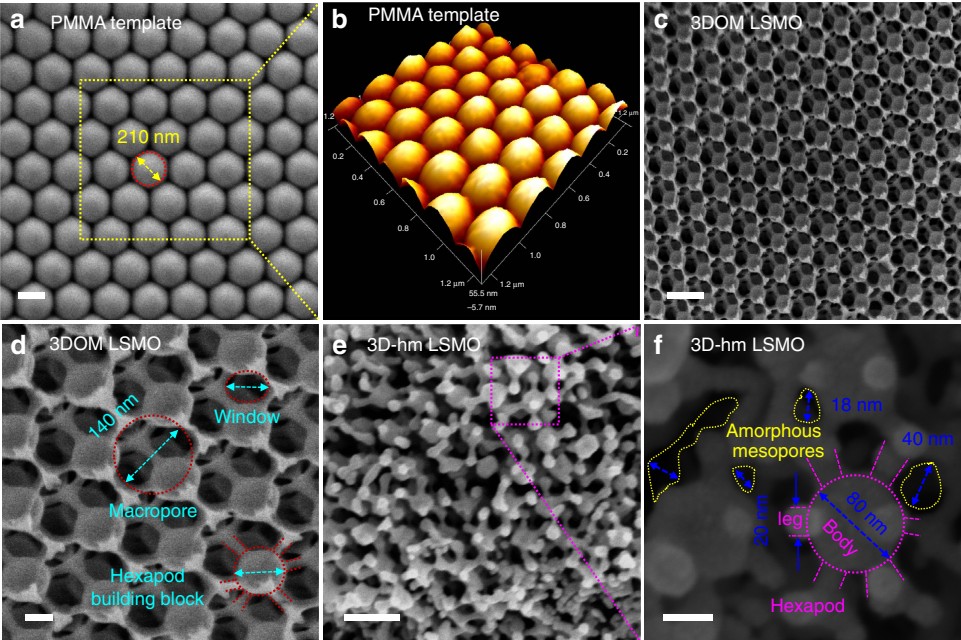

**Figure 2 | FE-HRSEM and 3D-eAFM analysis of LSMO catalysts.** (**a**) FE-HRSEM and (**b**) 3D-eAFM images of PMMA hard template, (**c,d**) FE-HRSEM images of 3DOM LSMO, (**e,f**) FE-HRSEM images of 3D-hm LSMO with structural parameters highlighted in different colours. Scale bars in **a,c,e** are 200 nm. Scale bars in **d,f** are 50 nm.

microspheres are interconnected and regularly oriented, forming a single-cubic crystal of 3DOM LSMO with L-lysine as the stabilizing surfactant (Fig. 3a,b). Intentional cleavage of the linkages confers direct access to hexapod-shaped nanoparticles (3D-hm LSMO) which survive intact from the sonication and centrifuge treatments (as shown in Fig. 3d − f and Supplementary Fig. 8). Two different representations of circular and quadrangle bodies with an inverse building block structure are shown in Supplementary Figs 9 and 10, respectively, which is in accordance with differences in the 3DOM architecture morphology depending on the cleavage points. According to the HAADF-STEM

image (Fig. 3c), the characteristic perovskite lattice fringes of the LSMO materials exhibit a $d$ spacing of 0.270 nm which is consistent with the (110) spacing of cubic $La_{0.6}Sr_{0.4}MnO_3$ perovskite, indicating the formation of 3DOM LSMO with good crystallinity (JCPDS PDF No. 04-016-6114). Figure 3d–f show the typical 3D-hm LSMO architecture comprising spherical pores which are interconnected via the neighbouring legs ($51 \times 43$ nm in length) through open windows (80 nm in diameter) with an interconnected macropore wall thickness of 13–19 nm, which agrees with the FE-HRSEM images (Fig. 2). The concave and convex curvatures on the surface of the hexapod unit are

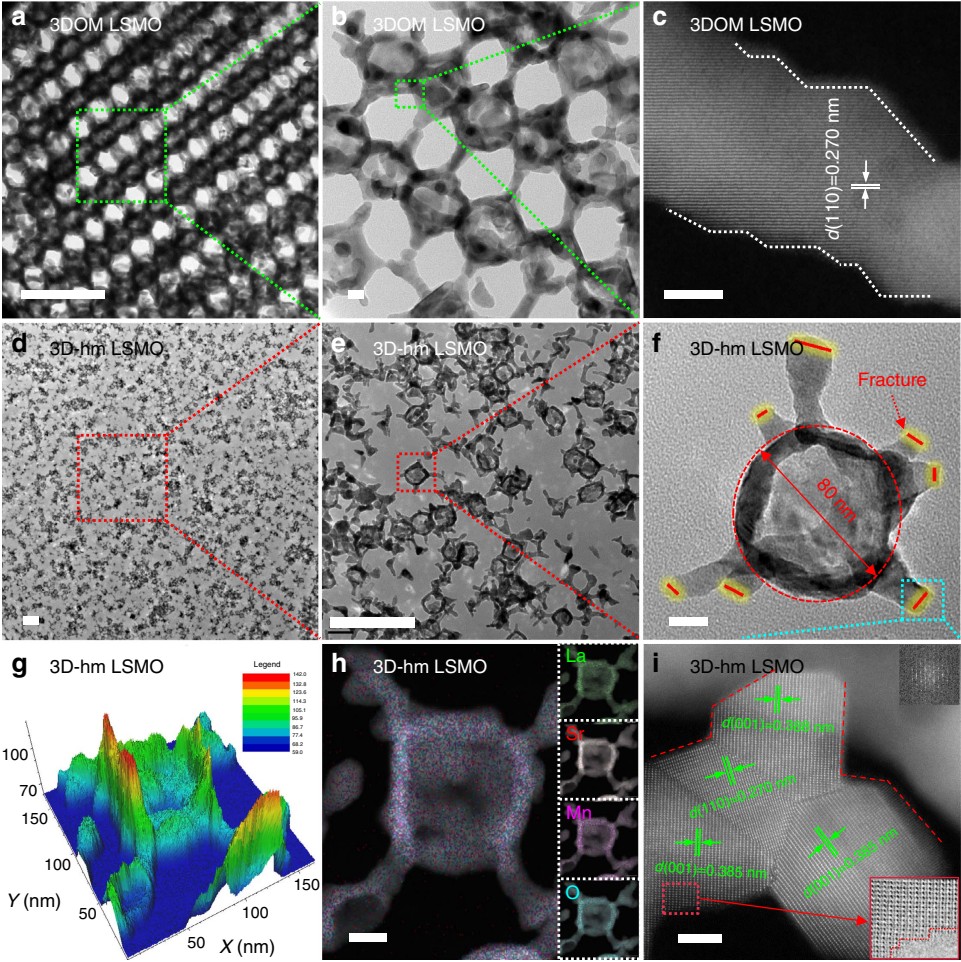

**Figure 3 | FE-HRTEM analysis of LSMO catalysts.** (**a,b**) FE-HRTEM images of 3DOM LSMO, (**c**) HAADF-STEM image of 3DOM LSMO, (**d–f**) FE-HRTEM images of 3D-hm LSMO, (**g**) 3D topography of 3D-hm LSMO extracted from (**h**) combined HAADF-STEM-EDS mapping of La, Sr, Mn and O with individual element mapping shown in the insets, and (**i**) HAADF-STEM image of the intersecting surface of the fractures in 3D-hm LSMO. Scale bars in **a,d,e** are 300 nm. Scale bars in **b,f,h** are 20 nm. Scale bars in **c,i** are 5 nm.

observed, as shown in the 3D topographical image (Fig. 3g). The four elements (La, Sr, Mn and O) are well dispersed in the structure, as revealed in the HAADF-STEM-EDS mapping image (Fig. 3h). Importantly, the high magnification image of the hexapod fracture line (Fig. 3i) indicates the exposure of a new (001) facet, as compared with the 3DOM LSMO which predominantly presents the (110) facet (Supplementary Figs 11 and 12). The multiple bright electron diffraction rings in the selected area electron diffraction pattern recorded for the sample shows the formation of a well-crystallized LSMO perovskite (inset of Fig. 3i).

XRD analysis indicates that the selected calcination conditions (750 °C for 4 h) are suitable to form a crystalline material possessing a cubic symmetry (space group: Pm-3m, lattice parameter (a): 0.3874 nm) which is comparable to the ideal LSMO perovskite-type structure (JCPDS PDF No. 04-016-6114)[6,9]. The XRD measurements also verify the crystal composition of 3D-hm LSMO is unchanged following the fragmentation treatment as it exhibits the same crystallographic parameters (cubic LSMO phase) as the 1DDN LSMO and 3DOM LSMO samples (Supplementary Fig. 13). $N_2$ adsorption–desorption isotherms of the 3D-hm and 3DOM LSMO samples show similar type IV isotherms (Supplementary Fig. 14a). The isotherms for both the 3D-hm LSMO and 3DOM LSMO samples (not the 1DDN LSMO sample) display H3- ($p/p_0 = 0.9$–$1.0$) and

H2- ($p/p_0 = 0.2$–$0.6$) type hysteresis loops, demonstrating the retention of macroporosity and mesoporosity within in the fragmented material (Supplementary Fig. 14b)[10]. Fragmentation of the 3DOM LSMO has a significant impact on its Brunauer-Emmett-Teller (BET) surface area and pore volume, increasing from $33.5\,m^2\,g^{-1}$ and $0.11\,cm^3\,g^{-1}$ for 3DOM LSMO to $48.9\,m^2\,g^{-1}$ and $0.15\,cm^3\,g^{-1}$ for 3D-hm LSMO (a 46% increase), respectively. The increase in surface area ($\sim 30\%$) arising from fragmentation is anticipated to derive primarily from the newly exposed (001) facet in the 3D-hm. In contrast, the BET surface area of the 1DDN LSMO is $4.3\,m^2\,g^{-1}$, which is more than 10 times lower than the 3D-hm LSMO surface area.

**Surface compositions.** XPS measurements were undertaken to identify any effect the fragmentation process may have had on surface atomic ratios and elemental state changes in the 3D-hm LSMO. The results are summarized in Table 1 and Supplementary Table 3. The results indicate that the surface La/Mn molar ratios (0.43–0.50) of all the LSMO samples are lower than the nominal La/Mn molar ratio (0.6), suggesting their surfaces are enriched with Mn. $Mn^{\delta +}$ species with a higher level of oxidation have been reported to be more active for $CH_4$ oxidation than both $Mn^{\delta +}$ species with a lower valence and metallic $Mn^0$ (ref. 9). The 3D-hm LSMO sample has the largest

**Table 1 | BET surface areas, ratios of oxygen species, catalytic activities, specific reaction rates and apparent $E_a$ of the 1DDN, 3DOM and 3D-hm LSMO samples.**

| Catalyst | BET surface area ($m^2 g^{-1}$) | $Mn^{4+}/Mn^{3+}$ molar ratio | Methane combustion activity (°C) | | | Specific reaction rate at 375 °C | | $E_a$ (kJ mol$^{-1}$) |
|---|---|---|---|---|---|---|---|---|
| | | | $T_{10\%}$* | $T_{50\%}$† | $T_{90\%}$‡ | $r_a$ (mol $g_{cat}^{-1} s^{-1}$) | $r_a'$ (mol $m^{-2} s^{-1}$) | |
| 1DDN LSMO | 4.32 | 1.13 | 402 | 480 | 690 | $1.50 \times 10^{-6}$ | $3.48 \times 10^{-7}$ | 136 |
| 3DOM LSMO | 33.5 | 1.44 | 342 | 385 | 510 | $1.18 \times 10^{-5}$ | $3.53 \times 10^{-7}$ | 102 |
| 3D-hm LSMO | 48.9 | 1.51 | 310 | 360 | 438 | $2.36 \times 10^{-5}$ | $4.84 \times 10^{-7}$ | 67.3 |

3DOM, three-dimensionally ordered macroporous; 1DDN, 1D nonporous; 3D-hm LSMO, 3D-hm La$_{0.6}$Sr$_{0.4}$MnO$_3$; $E_a$, activation energy.
*The temperatures required for 10% methane conversion.
†The temperatures required for 50% methane conversion.
‡The temperatures required for 90% methane conversion.

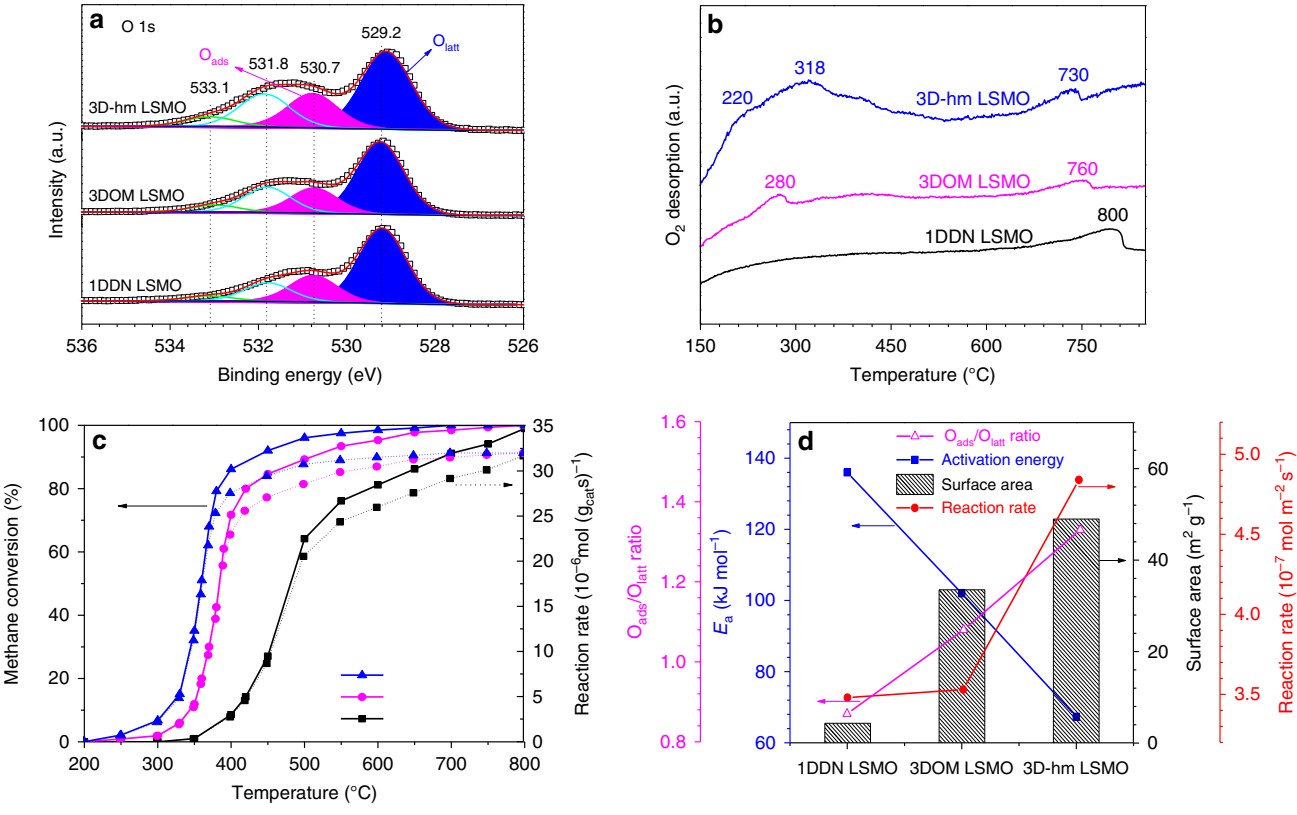

**Figure 4 | Oxygen species analysis and catalytic performance over LSMO catalysts.** (**a**) O 1s XPS spectra, (**b**) O$_2$-TPD profiles, (**c**) methane combustion activities for the 1DDN, 3DOM and 3D-hm LSMO samples, and (**d**) the association activation energy and reaction rate has with surface oxygen composition (O$_{ads}$/O$_{latt}$ ratio) and BET surface area for the LSMO samples.

surface area (48.9 m$^2$ g$^{-1}$) and the highest surface Mn$^{4+}$/Mn$^{3+}$ molar ratio (1.51), as shown in Table 1 and Supplementary Fig. 15. According to Fig. 4a, the fractions of the components in the O 1s spectra that are attributed to surface lattice O$^{2-}$ (O$_{latt}$), adsorbed oxygen (O$_{ads}$), hydroxyl and/or carbonate groups and adsorbed water species vary non-monotonically with the different LSMO morphologies. The O$_{ads}$/O$_{latt}$ molar ratio increases from 0.35 to 0.44 upon using the fragmentation protocol (Supplementary Table 3). The result is consistent with the O$_2$-temperature programmed desorption (TPD) result (Fig. 4b), which reveals two desorption peaks, one below and one above 400 °C, indicating the release of O$_{ads}$ and O$_{latt}$ species, respectively[11]. On comparing the 3D-hm and 3DOM LSMO samples, the 3D-hm LSMO possesses a much larger and broader O$_{ads}$ desorption peak (spanning 200 to 400 °C) as well as a shift in the O$_{latt}$ peak to a lower temperature (730 °C). The result demonstrates that the fragmentation process provides an LSMO surface enriched in O$_{ads}$ species and improves

the mobility of the O$_{latt}$ species. No obvious low-temperature desorption peak is apparent for the 1DDN LSMO as it likely possesses fewer O$_{ads}$ species on the surface. A higher O$_{ads}$ presence can facilitate the adsorption and activation of O$_2$ molecules and promote the performance of the catalyst for CH$_4$ oxidation[9,12].

**Catalytic performance in methane oxidation reaction.** From the CH$_4$ oxidation activity data (Fig. 4c and Table 1), the 3D-hm LSMO sample performs the best in terms of activity, giving $T_{10\%}$, $T_{50\%}$ and $T_{90\%}$ values of 310, 360 and 438 °C, respectively, which are lower than both the 1DDN LSMO and 3DOM LSMO catalysts. In addition, the 3D-hm LSMO was thermally and catalytically stable, as is shown in Supplementary Fig. 16, where CH$_4$ conversion was maintained at 650 °C for 50 h of on-stream reaction. Kinetic studies were performed across a temperature

range of $250 - 420\,^\circ\text{C}$ (Supplementary Fig. 17b), where the catalysis occurred in a kinetically controlled regime as conversions were $<15\%$. As shown in Table 1, the $E_a$ values for 1DDN ($136\,\text{kJ}\,\text{mol}^{-1}$) and 3DOM ($102\,\text{kJ}\,\text{mol}^{-1}$) LSMO are higher than for the 3D-hm LSMO sample ($67.3\,\text{kJ}\,\text{mol}^{-1}$). Figure 4d depicts the association reaction rate ($r'_a$) and apparent activation energy ($E_a$) has with the $\text{O}_{\text{ads}}/\text{O}_{\text{latt}}$ molar ratio and BET surface area of the LSMO samples. The catalytic performance decreases in the sequence of 3D-hm LSMO > 3DOM LSMO > 1DDN LSMO and coincides with increases in both the BET surface area and $\text{O}_{\text{ads}}/\text{O}_{\text{latt}}$ molar ratio of the materials. The higher surface area, the mesopore presence, the greater amount of $\text{O}_{\text{ads}}$ species and the improved $\text{O}_{\text{latt}}$, mobility exhibited by the 3D-hm LSMO combine to play important roles in the oxidation of $\text{CH}_4$. The 3D-hm LSMO provides a higher specific reaction rate compared to the pristine 3DOM LSMO on the basis of both mass (2.0 times improvement) and surface area (1.4 times improvement), indicating that the increased surface area arising from fragmentation is not the sole factor enhancing the 3D-hm LSMO catalytic performance.

The kinetic investigation demonstrates that the 3D-hm LSMO catalyst delivers a reaction rate higher than and an $E_a$ value lower than what has been previously reported for $\text{CH}_4$ oxidation (Supplementary Table 4). We believe that our unique 3DOM disassembly protocol enhances both the BET surface area to introduce additional active sites as well as enriches the chemisorbed oxygen species coverage which, in turn, facilitates catalytic performance. The enlarged surface area may be attributed to the increase in exposed surface which occurs upon fragmenting the 3DOM architecture. Importantly, fragmentation also prompts exposure of the (001) facet at the fractured faces of the 3D-hm LSMO (Fig. 3i), which may contribute to the increase in adsorbed oxygen species on the 3D-hm LSMO sample.

**Density functional theory studies.** To further understand the elevated activity exhibited by the 3D-hm LSMO catalyst, DFT calculations were employed to reveal the origin of the reduced activation barrier for $\text{CH}_4^* \rightarrow \text{CH}_3^* + \text{H}^*$. Slab models for the (001) and (110) surfaces of LSMO were used to represent the exposed facets of the 3D-hm structure (Supplementary Fig. 19). The surface formation or cleavage energy provides an indication of the stability and ease by which a surface is formed. DFT calculations on the perovskites revealed that the cleavage energy for the (001) surface ($1.12\,\text{J}\,\text{m}^{-2}$) is lower than for the (110) surface ($1.92\,\text{J}\,\text{m}^{-2}$). The numbers agree well with other calculation values obtained for LaMnO$_3$ (ref. 13) and imply that fracturing the interconnected 3DOM skeleton will expose a greater percentage of the (001) facet. Adsorption energy calculations indicate that the (001) surface has a greater tendency to bind $\text{CH}_4$ molecules than the (110) surface (details on the associated DFT calculations can be found in Supplementary Discussion). The Van der Waals corrected calculations show that the (001) $\text{MnO}_2$-terminated layer binds $\text{CH}_4$ more strongly than the (001) SrO-terminated layer. On the $\text{MnO}_2$-terminated slabs, $\text{CH}_4$ molecules are the most stable when they are adsorbed on a Mn site with a H atom being directed towards the surface ($E_b = -0.25\,\text{eV}$) followed by adsorption on an O site ($E_b = -0.18\,\text{eV}$) (calculated by Supplementary Equation 1). On the SrO-terminated layer, the binding is weaker and does not exceed $-0.14\,\text{eV}$ for adsorption on the O site. This is in contrast to the (110) surface which is characterized by an overall weak binding and is virtually unsuitable for $\text{CH}_4$ adsorption. Assorted stable structures are provided in Fig. 5. On the basis of an extended exploration of the stability of several $\text{CH}_4$ and $\text{CH}_3^* + \text{H}^*$ sites on both surfaces, the energy barriers needed for the reaction to proceed can be estimated (other dehydrogenation steps on the (001) surface are included in

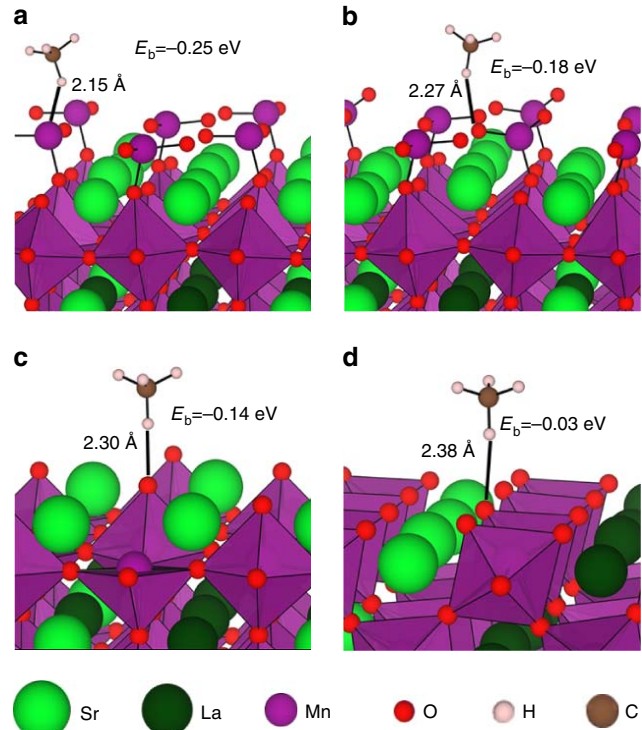

**Figure 5 | DFT analysis.** Stable adsorbed $\text{CH}_4$ configurations on the (001) and (110) surfaces: (**a**) $\text{CH}_4$ adsorbed on Mn and (**b**) on O on the (001) $\text{MnO}_2$-terminated surface, (**c**) $\text{CH}_4$ adsorbed on O on the (001) SrO-terminated surface and (**d**) the (110) surface.

Supplementary Fig. 20). Our calculation results reveal that the reaction is more facile on the (001) surface, where the barrier is calculated to be $1.04\,\text{eV}$, as compared to the (110) surface ($1.39\,\text{eV}$), a result consistent with the observed enhancement in catalytic methane oxidation.

## Discussion

An effective fragmentation strategy to generate a mesostructured 3D network composing of hexapod-shaped multi-metal oxide crystalline nanoparticles has been successfully exploited. The study has demonstrated that the fragmentation protocol invokes new structural and chemical features within the 3D-hm LSMO perovskite which elevate its catalytic activity for methane oxidation. The improved features include a modified mesoporous structure, a richness of surface oxygen species, better lattice oxygen mobility and exposure of the (001) facet along the fractured faces. DFT calculations indicated that the freshly exposed (001) surfaces of the 3D-hm LSMO impart a lower cleavage energy for the first $\text{C} - \text{H}$ bond in $\text{CH}_4$ compared to the (110) surface on the 3DOM LSMO. The findings highlight the potential for using structural disassembly as a means of synthesizing unique nanoscaled particles with versatile properties and opens exciting possibilities for attaining highly effective materials for catalytic systems.

## Methods
**Synthesis of 3DOM LSMO.** Two solvents, namely solvent A and solvent B, were prepared separately: solvent A comprised desired amounts of La(NO$_3$)$_3$.6H$_2$O, Sr(NO$_3$)$_2$, Mn(NO$_3$)$_2$, 3.0 ml ethanol, 3 ml ethylene glycol (EG) and 7.0 ml deionized water while solvent B comprised 0.7 g L-lysine, 0.4 g citric acid and 2.0 ml deionized water. After complete dissolution of each of the components, the two solvents were mixed together and stirred for 4 h to give a transparent solution. An aliquot of 2.0 g PMMA was added to the precursor solution and stirred for 4 h, followed by filtration and drying at 100 °C for 12 h. A two-step calcination process was utilized: the obtained solid was first calcined N$_2$ flow (90 ml per min) at a ramp of 1 °C per min from RT to 300 °C, where it was maintained for 3 h. The material

was then cooled to RT and calcined in air (90 ml per min) from RT to 750 °C at 1 °C per min where it was held for 4 h. 1DDN LSMO was prepared by a citrate method[14]. In brief, the metal nitrate precursors were dissolved in deionized water with equimolar citric acid. A porous gel was obtained by maintaining the nitrate liquid at 100 °C for 12 h after which it was calcined at a ramp rate of 5 °C per min from RT to 750 °C, where it was kept for 4 h.

**Synthesis of 3D-hm LSMO.** A 3D hexapod mesoporous LSMO perovskite structure was synthesized using a sequential assembly/disassembly strategy. Initially, 0.2 g of 3DOM LSMO was added to 10.0 ml of ethanol in a Teflon tube. The 3DOM LSMO suspension was then sonicated for repeat 10 min periods using either an ultrasonic probe (Misonix, Ultrasonic Liquid Processors) with 1 min resting intervals over a 2 h time-frame. During the fragmentation process, the suspension temperature was maintained below 25 °C by an ice bath. After sonication, the suspension was centrifuged for 10 min at 5,000 r.p.m. and 15 °C to isolate the supernatant which contained the LSMO hexapods and fragmented legs. The suspension was collected while the settled solids were redispersed, sonified and centrifuged again following the same procedure. As a final step, the collected supernatants were dried at 100 °C for 12 h to obtain the 3D-hm LSMO perovskites. The preparation details of PMMA and the comparison methods of 3D-hm LSMO are specified in Supplementary Methods.

**Catalytic activity evaluation.** Catalytic activity was assessed using a fixed-bed quartz tubular microreactor (i.d. = 6.0 mm) at atmospheric pressure for the complete oxidation of methane, as shown in Scheme S3. The catalyst sample (50 mg) was first pre-treated in a pure $O_2$ (30 ml per min) at 300 °C for 2 h and then in a $N_2$ (30 ml per min) as it cooled to RT. The reactant gas mixture containing 5 vol% $CH_4$ + 30 vol% $O_2$ + 65 vol% $N_2$ was then introduced to the reactor at a total flow of 42.8 ml per min, thus giving a gas hourly space velocity of ca. 50,000 ml $(g\,h)^{-1}$. The concentrations of the reactants and products in the reactor effluent were monitored on-line by a gas chromatograph (Young Lin 6500) equipped with a thermal conductivity detector (TCD) detector and a Carboxen-1010 PLOT column. The catalytic activities of the samples were evaluated using the temperatures $T_{10\%}$, $T_{50\%}$ and $T_{90\%}$ required for methane conversions of 10%, 50% and 90%, respectively. $CH_4$ conversion was determined using $(c_{inlet} - c_{outlet})/c_{inlet} \times 100\%$, where $c_{inlet}$ and $c_{outlet}$ are the $CH_4$ concentrations in the inlet and outlet feed streams, respectively.

**Characterization techniques.** XRD analyses were performed on a PANalytical Empyrean II diffractometer. SEM, TEM and HAADF-STEM images were collected on FEI Nova NanoSEM 450 FE-SEM microscope, Philips CM200 apparatus and JEOL JEM-ARM200F STEM, respectively. The specific surface areas and pore size distributions were obtained on a Micromeritics Tristar 3030 adsorption analyser and $O_2$-TPD analysis was on a Micromeritics Autochem II apparatus. XPS analyses were performed on a Thermo Scientific, UK (model ESCALAB250Xi). The operation conditions are listed in Supplementary Methods.

**Density functional theory calculations.** DFT, as implemented in Vienna ab initio simulation package (VASP)[15], was used to carry out the calculations presented here. The projector augmented wave (PAW)[16] method was used to treat the effective interaction of the core electrons and nucleus with the valence electrons while exchange and correlation were described using the Perdew-Burke-Ernzerhof (PBE) functional[17]. Plane waves with a kinetic energy cut-off of 500 eV were used to expand the Kohn–Sham wave-functions. Energies were converged during self-consistency to within $1 \times 10^{-5}$ eV tolerance, and forces were optimized until they were no more than $5 \times 10^{-2}$ eV Å$^{-1}$. K-points were sampled using $4 \times 4 \times 1$ and $4 \times 3 \times 1$ mesh. To improve the electronic description of LSMO used a Hubbard correction with $U_{Mn} = 3$ eV was used. The operation conditions are listed in Supplementary Discussion.

**Data availability.** The authors declare that all the other data supporting the findings of this study are available within the article and its Supplementary Information file or from the corresponding authors on reasonable request.

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

## Acknowledgements

Y.W. thanks the University International Postgraduate Award (UIPA) programme from The University of New South Wales for the Ph.D. Scholarship. H. A. acknowledges financial support through the Vice Chancellor Research Fellowship (RG142406) programme from the University of New South Wales. We acknowledge Drs Rhiannon Kuchel, Katie Levick, Yin Yao and Sean Lim from the UNSW Mark Wainwright Analytical Centre and Dr David Mitchell from the Electron Microscopy Centre, University of Wollongong for their generous help with the SEM, TEM, AFM and HAADF-STEM analyses. We acknowledge Dr. Mandalena Hermawan and Dr. Victor Wong from the UNSW for their help for laboratory management.

## Author contributions

Y.W. and H.A. performed catalyst synthesis, characterisation and activity testing. H.A.T. and X.T. conducted DFT calculations. J.S., H.D., J.D.G., A.L.R., S.C.S. and R.A. provided technical and scientific support as well as played a supervisory role in the research. The manuscript was prepared by Y.W., H.A., and H.A.T., and revised by all the authors.
