## [Peer Review File · Nature Communications]

Reviewers' comments:

Reviewer #1 (Remarks to the Author):

Recommendation: Major concerns must be addressed before considering the manuscript for publication.

The authors provide enough detail on the synthesis of the catalysts and perform very appropriate characterization of the materials, which supports their claims about generating stable fragments from 3DOM LSMO.

CH₄ oxidation is of current relevance for the scientific community given the increase in CH₄ production. Additionally, advanced catalyst design and synthesis is of great importance for the research community. Nevertheless, for publication in this top scientific journal, this reviewer considers the authors must address major concerns regarding the catalytic performance of the material.

Major concerns:

1. One of the main conclusions of the manuscript is that the 3D-hm LSMO performs better for CH₄ oxidation when compared to 3DOM LSMO and 1DDN LSMO, which is true under the perspective presented in Table 1. However, when the specific reaction rate is expressed per surface area of the catalyst, the 3D-hm LSMO has the lowest reaction rate (0.56 micromol/m²/s) when compared to the other two catalysts under study. Therefore, this reviewer is not convinced the increased performance of the 3D-hm LSMO lies beyond the greater surface area this catalyst offers due to the fragmentation procedure. Authors should comment on the reaction rates in units of micromol/m²/s.

2. Authors should describe in detail the way the number of active sites was quantified in order to report TOF in Table 1. It is not straightforward for TOF to be calculated for oxide catalysts.

3. The authors claim that the exposed (001) surfaces of the 3D-hm LSMO are responsible for the better performance of this catalyst because this surface favors the cleavage of the first C-H bond in CH₄. Nonetheless, this reviewer is not convinced that this step constitutes the rate determining step (RDS) for CH₄ oxidation, and would rather suspect CH₂* or CH* dehydrogenation to be the RDS. Did the authors perform calculations to confirm the first C-H bond breaking in CH₄ as the RDS?

Fig. S14 shows the potential energy diagram for the elementary step CH₄* ⇌ CH₃*+H*. However, one can observe two energetic barriers and three local minima on each path. What does this mean for a single elementary step? I think the authors can better describe this diagram.

Minor concerns:

1. Scheme 1 is not easy to follow.
2. Legend in Scheme 3 needs to be corrected.
3. Increase font size for FigS3 and FigS4

Reviewer #2 (Remarks to the Author):

The study focuses on the development of disassembly of mesostructured perovskite materials based on lanthanum, strontium, and manganite oxides. The reviewer found novelty and also much advances in the catalysis knowledge. Authors highlighted a novel fragmentation strategy for producing hexapod-shaped building block La_{0.6}Sr_{0.4}MnO₃ perovskite particles with rich surface oxygen species, modified mesoporous structure and exposure of the (001) facet along the fractured faces and examine the influence the surface chemistry and structure of the disassembled skeletons by XPS and DFT studies toward CH₄ combustion. The new exposed (110) facets by fragmentation strategy on the 3DOM LSMO surface enhanced methane oxidation activity. Given

the fact that the authors presented and argued their results in an interesting and consistent manner and the paper is also well written and organized properly, there is continued interest in the manuscript to be published in Nat. Commun. once it has been edited accordingly:

1. The mass transfer in a porous catalyst is one of the main factors contribute to the activity of the catalysts. It is important to estimate the influence of external and internal pore diffusion since the reaction rate can be affected by the diffusion limitation of the pore size of the catalysts. The authors have to consider this issue for the porous catalysts.

2. In Fig.2 c and i, the authors have highlighted the different configuration of the LSMO as terrace, step and kink. As my understanding, the critical point and change from "3DOM" to "3D-hm" is the facet but not the configuration. The authors also didn't make any discussion in the text about the configuration. What is the point to mark them in the TEM images? Is there any different such as generating more terrace, step or kink sites after the fragment treatment? If so, what is the effects of different configuration on the catalytic performance.

3. The stability investigation of 3D-hm LSMO should be completed. The reviewer agrees with the authors' conclusion about the excellent stability of 3DOM structure due to its mechanical strength which is not likely to agglomerate at high temperature. However, the nanoparticles generated from the fragment treatment seem to be lack of the mechanical strength. It is necessary to evaluate both activity and stability for the new generated catalysts.

4. The unit of TOF (s^{-1}) is not integrated. Normally the unit should be $mol/(g s)$ or $\mu mol/(g s)$ which mean per gram catalyst per second catalyze how many molar of the reactant.

Reviewer #3 (Remarks to the Author):

The manuscript fabricated a 3D hexapod mesoporous LSMO perovskite via a new fragmentation strategy. The 3DOM architecture, using as the precursor material, was disassembled into well-defined structural building units by a sonication method. The perovskites were characterized by means of XRD, FE-HRSEM, FE-HRTEM, N₂ adsorption-desorption, O₂-TPD and XPS and evaluated for methane oxidation. Although the perovskite with the special structure exhibited good catalytic performances, there still exist several questions. So, I do not think the manuscript can be published in Nature Communications. Here are some detail comments.

1. The innovation of the manuscript is not enough. Authors believed that it was the first time to synthesis perovskite with hexapod mesoporous structure. However, I think the special structure has always existed in the 3DOM structure. Put it another way, the 3DOM structure is composed of many interconnected hexapod building blocks. Via sonication method, only the combination of the original blocks will be broken.

2. Why only 70% of the 3DOM material can convert into 3D-hm particles. Can this proportion further increase through extending the ultrasonic time or some other methods?

3. The authors believed that a large area of (001) surface existed in the 3D-hn LSMO perovskite. However, it is not explained and discussed in terms of the characterization results. Is there any direct evidence for this phenomenon? And why (001) surface will generate just after sonicated for a while?

4. Few errors should be corrected. In the captions of Figure S1, S2 and S13, "EF-HRSEM" should be corrected as "FE-HRSEM".

The changes made to the original manuscript have been highlighted in **RED** for easy reference. Our responses are as follows:

Reviewer #1: Recommendation: Publish after major revisions noted.

The authors provide enough detail on the synthesis of the catalysts and perform very appropriate characterization of the materials, which supports their claims about generating stable fragments from 3DOM LSMO. CH₄ oxidation is of current relevance for the scientific community given the increase in CH₄ production. Additionally, advanced catalyst design and synthesis is of great importance for the research community. Nevertheless, for publication in this top scientific journal, this reviewer considers the authors must address major concerns regarding the catalytic performance of the material.

Comment 1: *One of the main conclusions of the manuscript is that the 3D-hm LSMO performs better for CH₄ oxidation when compared to 3DOM LSMO and 1DDN LSMO, which is true under the perspective presented in Table 1. However, when the specific reaction rate is expressed per surface area of the catalyst, the 3D-hm LSMO has the lowest reaction rate (0.56 micromol/m²/s) when compared to the other two catalysts under study. Therefore, this reviewer is not convinced the increased performance of the 3D-hm LSMO lies beyond the greater surface area this catalyst offers due to the fragmentation procedure. Authors should comment on the reaction rates in units of micromol/m²/s.*

Response 1: We thank reviewer 1 for the important observation. On re-evaluating the specific reaction rate, 400°C is not the best temperature to illustrate the improvement in catalytic performance introduced by fragmentation of the 3DOM LSMO.

Re-visiting Figure 3(C) in the manuscript indicates the positive contribution to reaction rate arising from fragmentation is more evident at temperatures below 400°C. This fact is highlighted in Figure R1 and Table R1 below. On comparing the performance of 3DOM LSMO and 3D-hm LSMO (i.e. the fragmented form of 3DOM LSMO) at temperatures of 350°C, 375°C and 400°C in terms of the specific reaction rate based on mass (r_a , mol/(g_{cat} s)) or surface area (r_a' , mol/(m² s)) of the catalyst, it is apparent that at 350°C and 375°C the fragmented catalyst outperforms the pristine 3DOM LSMO in all cases. At 375°C the fragmented 3DOM LSMO exhibits a 2.0 and 1.4 times greater reaction rate than the pristine 3DOM LSMO on the basis of mass and surface area, respectively. At 350°C the improvements are in the order of 2.9 and 2.0 times, respectively. The reaction rate and TOF at 375°C have now been included in Table 1 in the revised manuscript in place of the original

400°C values to more correctly reflect the beneficial impact of the newly exposed (001) facet on the 3D-hm LSMO on catalytic methane oxidation.

Consequently, the increase in surface area invoked by fragmentation is not the sole contributor providing to the better performance. The experimentally determined lower apparent activation energy (E_a) of 3D-hm LSMO (77.9 kJ/mol, 18% lower than that of 3DOM LSMO) in conjunction with the findings from the DFT study (i.e. the newly exposed (001) facet provides a reduced energy barrier for hydrogen abstraction) account for the improved performance by the fragmented material beyond the increase in specific surface area. The original 400°C reaction rate comparison in Table 1 (and associated discussion on page 11) has now been changed to be a comparison at 375°C in the modified manuscript to better reflect the impact of fragmentation on catalyst performance. Additionally, the calculation steps for evaluating reaction rate based on catalyst mass and surface area are given in section 6 of the revised Supporting Information.

Table R1. Specific reaction rate of LSMO samples at 350, 375 and 400 °C on the basis of catalyst mass and catalyst surface area.

Catalyst	Specific reaction rate at 350 °C		Specific reaction rate at 375 °C		Specific reaction rate at 400 °C	
	r_a [mol/(g _{cat} s)]	r'_a [mol/(m ² s)]	r_a [mol/(g _{cat} s)]	r'_a [mol/(m ² s)]	r_a [mol/(g _{cat} s)]	r'_a [mol/(m ² s)]
1DDN LSMO	3.25×10^{-7}	7.53×10^{-8}	1.50×10^{-6}	3.48×10^{-7}	2.74×10^{-6}	6.35×10^{-7}
3DOM LSMO	3.82×10^{-6}	1.14×10^{-7}	1.18×10^{-5}	3.53×10^{-7}	2.30×10^{-5}	6.85×10^{-7}
3D-hm LSMO	1.12×10^{-5}	2.30×10^{-7}	2.36×10^{-5}	4.84×10^{-7}	2.76×10^{-5}	5.64×10^{-7}

Figure R1. Methane conversion and reaction rate of LSMO samples versus temperature.

Comment 2: Authors should describe in detail the way the number of active sites was quantified in order to report TOF in Table 1. It is not straightforward for TOF to be calculated for oxide catalysts.

Response 2: As highlighted by reviewer 1, it is challenging to accurately calculate TOF values for oxide catalysts in heterogeneous catalysis, although it is possible to compare the intrinsic catalytic rate of different metal oxides by utilising a uniform reaction model. It is difficult to determine the exact number of active sites and, most importantly, the catalytically active sites may change during the reaction (e.g. due to agglomeration and re-dispersion). In our study we calculated the TOF value of each sample on the basis of mol CH₄ converted/(mol LSMO catalyst.s). The units defining TOF has now been corrected (and expanded as a tabular footnote) in Table 1 of the revised manuscript. Details on the calculations used to estimate the TOFs are now listed in section 6 of the revised Supporting Information.

Comment 3: The authors claim that the exposed (001) surfaces of the 3D-hm LSMO are responsible for the better performance of this catalyst because this surface favors the cleavage of the first C-H bond in CH₄. Nonetheless, this reviewer is not convinced that this step constitutes the rate determining step (RDS) for CH₄ oxidation, and would rather suspect CH₂ or CH* dehydrogenation to be the RDS. Did the authors perform calculations to confirm the first C-H bond*

breaking in CH₄ as the RDS? Fig. S14 shows the potential energy diagram for the elementary step CH₄ → CH₃*+H*. However, one can observe two energetic barriers and three local minima on each path. What does this mean for a single elementary step? I think the authors can better describe this diagram.*

Response 3: While the rate determining step is dependent on the details of the surface, many studies have shown that the first dehydrogenation step is the RDS. This formed the basis of our initial assumption, for which we compared this step on the (001) and (110) LSMO surfaces. Based on the query by reviewer 1 we performed additional calculations for the successive dehydrogenation steps focusing on the (001) surface and found that first step is truly the rate determining step (see Figure R2 below). The CH₃ and CH₂ dehydrogenations have comparable but slightly lower barriers than the CH₄ dehydrogenation. Therefore, our conclusions regarding the activity of the (001) surface compared to the (110) surface do not change.

The local minima that appear in Fig. S15 in the supporting information section of the original manuscript correspond to metastable sites on the LSMO surface. This arises due to the complexity of the potential energy landscape on the perovskite surface. In essence the NEB algorithm is able to bring one of the images to the saddle point and thus identify the correct transition state as shown in the figure.

Figure R2 Schematic profile of the energetics of the successive CH₄ dehydrogenation steps on the (001) LSMO surface. The activation barrier is shown in parenthesis.

The calculations confirming the rate determining step have now been provided in section 5 of the supporting information.

Minor concern 1. Scheme 1 is not easy to follow.

Response 1: Scheme 1 has been redesigned to be easier to follow. That is, the schematic diagram has been arranged as a flow diagram according to the preparation process as PMMA microsphere filled with precursor → 3DOM LSMO → inner connected hexapod LSMO, along with

SEM/TEM images of the PMMA, 3DOM LSMO and 3D-hm LSMO as additional illustration as follows:

Revised Scheme 1. Schematic illustration depicting synthesis of the 3D-hm LSMO catalysts.

Minor concern 2. Legend in Scheme 3 needs to be corrected.

Response 2: The legend in Scheme S3 has been modified accordingly in the revised Supporting Information. The confusing terms “Digital signal” and “Materials flow” have been removed from the original Scheme S3.

Minor concern 3. Increase font size for FigS3 and FigS4

Response 3: The font size for Fig S3 and Fig S4 have been increased from 14 to 26 in the revised Supporting Information section.

Reviewer #2: Recommendation: Publish after revisions noted.

The study focuses on the development of disassembly of mesostructured perovskite materials based on lanthanum, strontium, and manganese oxides. The reviewer found novelty and also much advances in the catalysis knowledge. Authors highlighted a novel fragmentation strategy for producing hexapod-shaped building block $\text{La}_{0.6}\text{Sr}_{0.4}\text{MnO}_3$ perovskite particles with rich surface oxygen species, modified mesoporous structure and exposure of the (001) facet along the fractured faces and examine the influence the surface chemistry and structure of the disassembled skeletons by XPS and DFT studies toward CH_4 combustion. The new exposed (110) facets by fragmentation strategy on the 3DOM LSMO surface enhanced methane oxidation activity. Given the fact that the authors presented

and argued their results in an interesting and consistent manner and the paper is also well written and organized properly, there is continued interest in the manuscript to be published in *Nat. Commun.* once it has been edited accordingly:

Comment 1: *The mass transfer in a porous catalyst is one of the main factors contribute to the activity of the catalysts. It is important to estimate the influence of external and internal pore diffusion since the reaction rate can be affected by the diffusion limitation of the pore size of the catalysts. The authors have to consider this issue for the porous catalysts.*

Response 1: Conducting a Weisz–Prater analysis is a means by which the presence of internal and external mass transfer limitations in a porous catalyst can be evaluated (*Energy Fuels*, 2009, 23: 86–93). A Weisz-Prater analysis for the 3D-hm LSMO catalyst at a CH₄ conversion of 50% and temperature of 360 °C with a GHSV of 50 000 mL/(g h) has now been included in the revised Supporting Information. According to the dimensionless Weisz-Prater (Nw-p) criterion, when the value of

$$N_{W-p} = \frac{r_a \cdot \rho_c \cdot R_p^2}{C_s \cdot D_{eff}} \leq 1$$
, internal mass transfer effects can be neglected and if

$$\frac{r_a \cdot \rho_b \cdot R_p \cdot n}{k_c \cdot C_{sb}} \leq 0.15$$
, then external mass transfer effects can be neglected. In our calculations, both of these values (5.2×10^{-9} and 7.9×10^{-8} , respectively) are well below the limiting values (1 and 0.15, respectively) which indicate mass transfer limitations due to pore diffusion are insignificant in the 3D-hm LSMO system. Details on the parameters used in Weisz-Prater analysis have been listed in Table S1 in section 7 of the revised Supporting Information.

Comment 2: *In Fig.2 c and i, the authors have highlighted the different configuration of the LSMO as terrace, step and kink. As my understanding, the critical point and change from “3DOM” to “3D-hm” is the facet but not the configuration. The authors also didn’t make any discussion in the text about the configuration. What is the point to mark them in the TEM images? Is there any different such as generating more terrace, step or kink sites after the fragment treatment? If so, what is the effects of different configuration on the catalytic performance.*

Response 2: Highlighting of the terrace, step and kink in the Fig. 2c and 2i HR-TEM images has now been removed as identifying the newly exposed facets is the important finding in our work, as correctly pointed out by reviewer 2. Other studies have previously highlighted the importance of co-ordinatively unsaturated sites on a catalyst surface such as steps, kinks, corners and edges and that they can provide active sites for heterogeneous catalysis. For example, Alex et al. utilised STM to illustrate that the surface steps of Au act as catalytic sites for Br atom removal

from the dibromoterfluorene molecule and consequently the polymerization process, leading to efficient molecular activation in contrast to the flat Au terraces (*Angewandte Chemie*, 2012, 51: 5096-5100). However, in our catalytic system there is no evidence showing a distinct variation in the density of kink, terrace and step sites upon fragmentation of the 3DOM structure. The predominant change upon fragmentation is exposure of the (001) facet.

Comment 3: *The stability investigation of 3D-hm LSMO should be completed. The reviewer agrees with the authors' conclusion about the excellent stability of 3DOM structure due to its mechanical strength which is not likely to agglomerate at high temperature. However, the nanoparticles generated from the fragment treatment seem to be lack of the mechanical strength. It is necessary to evaluate both activity and stability for the new generated catalysts.*

Response 3: In line with the suggestion by reviewer 2, a long-term stability test was conducted with the result now added as Fig. S13 in the revised Supporting Information section and related discussion included on page 10 of the revised manuscript. The stability test showed that the 3D-hm LSMO sample exhibits excellent thermal-stability at 650 °C for 50 h on-stream reaction.

Comment 4: *The unit of TOF (s⁻¹) is not integrated. Normally the unit should be mol/(g s) or μmol/(g s) which mean per gram catalyst per second catalyze how many molar of the reactant.*

Response 4: The TOFs values for the catalysts have been re-calculated using the suggested unit mol/(mol s) based on the moles of the catalyst, as detailed in Table 1 of the revised manuscript. Calculations for the TOFs and reaction rates have been also included in the revised Supporting Information section.

Reviewer #3: *The manuscript fabricated a 3D hexapod mesoporous LSMO perovskite via a new fragmentation strategy. The 3DOM architecture, using as the precursor material, was disassembled into well-defined structural building units by a sonication method. The perovskites were characterized by means of XRD, FE-HRSEM, FE-HRTEM, N₂ adsorption-desorption, O₂-TPD and XPS and evaluated for methane oxidation. Although the perovskite with the special structure exhibited good catalytic performances, there still exist several questions. So, I do not think the manuscript can be published in Nature Communications. Here are some detail comments.*

Comment 1: The innovation of the manuscript is not enough. Authors believed that it was the first time to synthesis perovskite with hexapod mesoporous structure. However, I think the special structure has always existed in the 3DOM structure. Put it another way, the 3DOM structure is composed of many interconnected hexapod building blocks. Via sonication method, only the combination of the original blocks will be broken.

Response 1:

We believe the innovation is sufficient for publication of the work in Nature Communications. We agree with the reviewer in that the 3DOM comprises many interconnected hexapod building blocks, and sonication liberates them from the structure.

However, the key innovations of this work lie in fragmenting the 3DOM framework to expose the (001) facet within the perovskite hexapods and identifying how exposing this facet promotes the catalytic activity of the perovskite beyond the benefit of a simple increase in surface area. That is, complementary to the increase in the surface area (i.e. kinetic improvement), our experiments and calculations demonstrate that cleaving the 3DOM structure along the (001) facet exposes a new crystal plane which lowers the energy required (i.e. thermodynamic improvement) to extract hydrogen from the methane molecule during reaction.

Comment 2: Why only 70% of the 3DOM material can convert into 3D-hm particles. Can this proportion further increase through extending the ultrasonic time or some other methods?

Response 2: The 70% refers to the yield (by weight) of the 3D-hm particles relative to the initial amount of 3DOM subjected to fragmentation (i.e. losses during the fragmentation process) and not the percentage of 3DOM converted to 3D-hm. Losses are likely to have occurred during the centrifuging/drying stages due to the fine size of the hexapod particles. The sentence in section 1.3 of the supporting information detailing this point has now been rewritten so as to clarify this.

Comment 3: The authors believed that a large area of (001) surface existed in the 3D-hn LSMO perovskite. However, it is not explained and discussed in terms of the characterization results. Is there any direct evidence for this phenomenon? And why (001) surface will generate just after sonicated for a while?

Response 3: In its original (i.e. non-fragmented) state the 3DOM architecture presents an exposed (110) facet. Only upon fragmentation of the structure does the (001) facet become exposed and this is because sonication leads to the crystal cleaving along this particular face. Consequently, the hexapods comprise both the (110) facet (the original 3DOM exposed facet) and the (001) facet (the newly exposed facet along which the crystal has been cleaved). HAADF-STEM provides direct evidence of

the (001) facet on the newly exposed faces of the 3D-hm particles (Figure 2 in the original manuscript). We have also included additional STEM images of the 3D-hm LSMO in the revised Supporting Information section (Fig. S8 and S9), which further confirm the (110) facet on the hexapod body and the (001) facet on cleaved arms of the hexapod blocks.

The short sonication process does not result in the generation of a new (001) facet in the 3D-hm crystal as indicated by the XRD profiles (Fig. S10 in Supporting Information) which show little variation in the (001) and (110) reflections of the 3DOM and 3D-hm materials. This observation is mentioned across pages 10/11 of the manuscript. The sonication process acts to cleave the 3DOM architecture along the (001) facet leading to its exposure in the 3D-hm hexapods.

The increase in surface area of the 3D-hm sample relative to the original 3DOM sample (i.e. 48.9 m²/g vs 33.5 m²/g) is anticipated to be mostly due to the newly exposed (001) facet meaning approximately 15 m²/g (i.e. ~ 30%) of the 3D-hm surface is comprised of the (001) facet. This has now been mentioned on page 9 on the revised manuscript.

Comment 4: Few errors should be corrected. In the captions of Figure S1, S2 and S13, “EF-HRSEM” should be corrected as “FE-HRSEM”.

Response 4: The above error in the Figs S1, S2 and S3 captions has now been corrected.

Reviewers' comments:

Reviewer #1 (Remarks to the Author):

The authors have made considerable effort to improve the manuscript and many of my concerns have been addressed. However, this reviewer considers few matters should be clarified before the manuscript is accepted for publication:

1. Are the activation energies reported in Fig 3D calculated under differential conditions (conversion no greater than 15 %)?
2. TOF was calculated as [moles of CH₄ reacted/mol catalyst/second]. However, this definition of TOF is not accurate, since TOF refers to [molecules of CH₄ reacted/active sites/second], where the active sites are only superficial sites without including atoms in the bulk of the catalyst. I recommend the authors refrain from defining TOF wrongly. To convey the main message of the paper, reporting the activity using Rates as mol/m²/s seems sufficient.
3. The variable r₂ is defined as 'conversion rate', do the authors mean only 'conversion'? The term 'conversion rate' does not seem appropriate.
4. They state :
ra = r₁/m, but m is defined as m²/g. The units do not match.
ra' = ra/s, but s is not defined in Table S1.
5. Did the authors confirm the transition state structures calculating the vibrational frequencies and verifying they are imaginary? If not, did the authors perform the CI-NEB calculations taking as the new initial state the metastable adsorption site? This fine-tuning of the calculations might alter the rate-determining step (RDS), given that the differences in activation barriers of the CH₄, CH₃ and CH₂ dehydrogenation steps are within DFT errors (0.1 eV).

If still the barriers do not change after fine-tuning the calculations, authors should not claim the first dehydrogenation step is the only RDS, but rather disclose the probable coexistence of multiple RDSs.

Reviewer #2 (Remarks to the Author):

I am convinced that the points raised in the previous round of review have been satisfactorily addressed.

Reviewer #3 (Remarks to the Author):

All the questions and comments were considered and addressed by the authors. The manuscript was improved and it can be accepted for the publication in Nature Communications.

Reviewer #1: *The authors have made considerable effort to improve the manuscript and many of my concerns have been addressed. However, this reviewer considers few matters should be clarified before the manuscript is accepted for publication.*

Comment 1: *Are the activation energies reported in Fig 3D calculated under differential conditions (conversion no greater than 15 %)?*

Response 1: The activation energies reported in Fig. 3D were originally calculated at 20% conversion values. As suggested by Reviewer 1, the activation energies have been recalculated at 15% conversion with the adjusted values (see Table R1 below) now presented in Table 1 and Fig. 3D (plus associated discussion on page 9) of the revised manuscript. As shown in Table R1 the E_a values for 1DDN LSMO and 3DOM LSMO at 15% conversion increase slightly (by ~8%) while the E_a value for 3D-hm LSMO decreases (by ~14%) when compared with the E_a values at 20% conversion. The E_a values at 15% conversion show the 3D-hm LSMO performs even better than the other catalysts (when compared to the E_a values at 20% conversion). More importantly though, the trend in E_a values for the three catalyst types remains the same for both conversions such that the findings of the work (and associated discussion) remain unchanged.

Table R1. Activation energies of LSMO samples calculated at 20% and 15% methane conversions.

Catalyst	Activation Energy E_a (kJ/mol)	
	Previous (20% conversion)	Adjusted (15% conversion)
1DDN LSMO	125	136
3DOM LSMO	94.7	102

3D-hm LSMO	77.9	67.3
------------	------	------

Comment 2: *TOF was calculated as [moles of CH₄ reacted/mol catalyst/second]. However, this definition of TOF is not accurate, since TOF refers to [molecules of CH₄ reacted/active sites/second], where the active sites are only superficial sites without including atoms in the bulk of the catalyst. I recommend the authors refrain from defining TOF wrongly. To convey the main message of the paper, reporting the activity using Rates as mol/m²/s seems sufficient.*

Response 2: According to the suggestion by Reviewer 1, the TOF values have been removed from Table 1 as well as related discussion within the revised manuscript. Additionally, any reference to the TOF in Fig. 3D has been changed to be reaction rate (mol/m² s) in the revised manuscript. The TOF calculations have also been removed from section 5 in the revised Supplementary Information.

Comment 3: *The variable r₂ is defined as ‘conversion rate’, do the authors mean only ‘conversion’? The term ‘conversion rate’ does not seem appropriate.*

Response 3: The definition of r₂ has been corrected to be conversion of CH₄ in Table S1 of the revised Supplementary Information.

Comment 4: *They state: $r_a = r_1/m$, but m is defined as m²/g. The units do not match. $r_a = r_a/s$, but s is not defined in Table S1.*

Response 4: We apologise as the confusion derived from a typographical error in that ‘m’ (m²/g, catalyst surface area) in Table S1 was meant to be ‘s’ while ‘m’ in the equation $r_a = r_1/m$ was meant to be ‘m’ (g, catalyst mass). The ‘s’ has now been corrected in Table S1 of the revised Supplementary Information.

Comment 5: *Did the authors confirm the transition state structures calculating the vibrational frequencies and verifying they are imaginary? If not, did the authors perform the CI-NEB calculations taking as the new initial state the metastable adsorption site? This fine-tuning of the calculations might alter the rate-determining step (RDS), given that the differences in activation barriers of the CH₄, CH₃ and CH₂ dehydrogenation steps are within DFT errors (0.1 eV). If still the barriers do not change after fine-tuning the calculations, authors should not claim the first dehydrogenation step is the only RDS, but rather disclose the probable coexistence of multiple RDSs.*

Response 5: While we did not explicitly calculate the vibrational frequencies of the transition state structures, at an earlier stage of the calculations we did explore the pathway

starting with the metastable adsorption site as the new initial state for our CI-NEB. This metastable site is about 0.1 eV higher than the original initial state and therefore one would expect that the activation barrier should decrease by ~ 0.1 eV (1.04 eV \rightarrow 0.94 eV). The fine-tuned value of the actual calculation turns out to be about 1.01 eV.

This value is only marginally higher than the activation barriers required for the second dehydrogenation step of the methane molecule. We therefore agree with Reviewer 1 in that the calculations suggest the possible coexistence of multiple RDSs. Related text has been modified on page 10 of the revised manuscript and section 4 of the revised Supplementary Information.

Reviewer #2: (Remarks to the Author):

I am convinced that the points raised in the previous round of review have been satisfactorily addressed.

Reviewer #3: (Remarks to the Author):

All the questions and comments were considered and addressed by the authors. The manuscript was improved and it can be accepted for the publication in Nature Communications.

REVIEWERS' COMMENTS:

Reviewer #1 (Remarks to the Author):

The authors have addressed properly all the comments. The manuscript is recommended for acceptance.